# Gestational Weight Gain Relates to DNA Methylation in Umbilical Cord, Which, In Turn, Associates with Offspring Obesity-Related Parameters

**DOI:** 10.3390/nu15143175

**Published:** 2023-07-17

**Authors:** Berta Mas-Parés, Sílvia Xargay-Torrent, Ariadna Gómez-Vilarrubla, Gemma Carreras-Badosa, Anna Prats-Puig, Francis De Zegher, Lourdes Ibáñez, Judit Bassols, Abel López-Bermejo

**Affiliations:** 1Pediatric Endocrinology Research Group, (Girona Biomedical Research Institute) IDIBGI, 17190 Salt, Spain; bmas@idibgi.org (B.M.-P.);; 2Materno-Fetal Metabolic Research Group, (Girona Biomedical Research Institute) IDIBGI, 17190 Salt, Spain; 3University School of Health and Sport (EUSES), University of Girona, 17190 Salt, Spain; 4Department of Development & Regeneration, University of Leuven, 3000 Leuven, Belgium; 5Endocrinology Department, Research Institute Sant Joan de Déu, University of Barcelona, 08950 Esplugues, Spain; 6Centro de Investigación Biomédica en Red de Diabetes y Enfermedades Metabólicas Asociadas (CIBERDEM), ISCIII, 28029 Madrid, Spain; 7Department of Pediatrics, Dr. Josep Trueta Hospital, 17007 Girona, Spain; 8Department of Medical Sciences, University of Girona, 17003 Girona, Spain

**Keywords:** DNA methylation, obesity, gestational weight gain

## Abstract

Excessive gestational weight gain (GWG) has a negative impact on offspring’s health. Epigenetic modifications mediate these associations by causing changes in gene expression. We studied the association between GWG and DNA methylation in umbilical cord tissue; and determined whether the DNA methylation and the expression of corresponding annotated genes were associated with obesity-related parameters in offspring at 6 years of age. The methylated CpG sites (CpGs) associated with GWG were identified in umbilical cord tissue by genome-wide DNA methylation (*n* = 24). Twelve top CpGs were validated in a wider sample by pyrosequencing (*n* = 87), and the expression of their 5 annotated genes (*SETD8*, *TMEM214*, *SLIT3*, *RPTOR*, and *HOXC8*) was assessed by RT-PCR. Pyrosequencing results validated the association of SETD8, SLIT3, and RPTOR methylation with GWG and showed that higher levels of *SETD8* and *RPTOR* methylation and lower levels of *SLIT3* methylation relate to a higher risk of obesity in the offspring. The association of *SETD8* and *SLIT3* gene expression with offspring outcomes paralleled the association of methylation levels in opposite directions. Epigenetic changes in the umbilical cord tissue could explain, in part, the relationship between GWG and offspring obesity risk and be early biomarkers for the prevention of overweight and obesity in childhood.

## 1. Introduction

There is a convincing association between pre and postnatal environmental exposures and disease risk in later life [1,2]. Maternal obesity, known to cause an unfavorable intrauterine environment, increases the offspring’s risk of chronic diseases, such as obesity and metabolic syndrome, in childhood and adulthood [1,3]. There is also evidence that high gestational weight gain (GWG) associates with body mass index (BMI) and overweight in children and adults, having a negative impact on long-term cardio-metabolic health [4,5,6].

Numerous studies have pointed out epigenetics as being the link between these environmental exposures and the offspring outcomes [1,7,8,9]. Epigenetic marks, including DNA methylation, can be changed under an unfavorable intrauterine environment [10] and are able to modulate gene transcription [11]. Alterations of the epigenome as a result of obesity have been broadly described in non-pregnant populations, studying both peripheral blood and adipose tissue [12]. Reports comparing changes of the methylome associated with maternal obesity in fetal tissues, such as umbilical cord blood, umbilical cord tissue, and placenta, are mostly focused exclusively on specific genes or on identifying changes related to pre-pregnancy BMI rather than GWG [13,14,15]; furthermore, most studies to date do not include long-term offspring outcomes [16]. For instance, Lesseur et al. described that an unfavorable perinatal environment (such as maternal obesity or gestational diabetes) could influence placental DNA methylation patterns in the leptin promoter [17,18], and Breton et al. found that DNA methylation of the *NEGR1* gene in the placenta could be related to BMI and neurodevelopment of the offspring [19]. As for whole methylome studies, Shrestha et al. performed an epigenome-wide association study (EWAS) in placental tissue and found trimester-specific changes in the DNA methylation profile that were associated with pre-pregnancy BMI and gestational weight gain [20]. Another genome-wide methylation study in the placenta showed that inadequate GWG causes abnormal DNA methylation, mostly at cytosine and guanine dinucleotides (CpG) island sites located at the promoter region of genes encoding transcriptional factors [21].

Fewer epigenetic studies have been performed in umbilical cord tissue. Thakali et al. used Bisulfite Amplicon Sequencing (BSAS) to conduct targeted DNA methylation association analysis of maternal obesity and excessive GWG with DNA methylation of selected genes related to metabolism and imprinted genes in the umbilical cord tissue [22]. Similarly, Chavira-Suárez et al. used the methylation-sensitive high-resolution melting (MS-HRM) technique and Sanger allele-bisulfite sequencing to examine the association of pre-pregnancy BMI and GWG with DNA methylation in selected obesogenic genes in umbilical vein tissue [8].

To our knowledge, no studies have examined, in umbilical cord tissue, the association between DNA methylation patterns related to GWG and offspring cardio-metabolic outcomes.

The umbilical cord tissue is an accessible tissue at birth that is composed of fetal cells. It is known that the umbilical cord is affected by the inutero environment, as its cells have a well-known differentiation potential and plasticity that can be disturbed by pregnancy-induced changes [23]. In early gestation, fetal organogenesis occurs, and the fetal epigenome is susceptible to environmental stimuli [20], which may condition biological responses, and in turn, can define the offspring’s disease risk later in life [1]. This tissue has been broadly used in epigenetics studies [22,24].

We hypothesize that umbilical cord methylation may be affected by GWG and, in turn, may associate with an adverse metabolic phenotype in the offspring. In this context, our objectives were: (1) to study the association between GWG and DNA methylation in umbilical cord tissue in a prenatal cohort of mother-infant pairs, (2) to determine whether the DNA methylation and expression of the identified genes associated with obesity-related parameters in the offspring at 6 years of age.

## 2. Materials and Methods

### 2.1. Subjects and Samples

This is a prospective longitudinal study of 111 healthy pregnant Caucasian women and their newborns, who were included in a mother-children cohort in Girona, north-eastern Spain. Women were recruited in a prenatal primary care setting during the first trimester of gestation and followed up until delivery, and clinical exams and blood tests were performed. Inclusion criteria were uncomplicated pregnancies and delivering infants at term (37 to 40 weeks). Exclusion criteria were women with major medical, surgical, or obstetrical complications, such as multiple pregnancies, hypertension, gestational diabetes or preeclampsia, fetal growth restriction, malformations, or asphyxia, as well as women whose pregnancy required assisted reproduction techniques. A subset of children whose families consented (70%) were followed up at age 6 years. The protocol was approved by the Institutional Review Board of Dr. Josep Trueta Hospital, and informed written consent was obtained from all parents.

To perform the study, 24 mother-infant pairs were used for the screening analysis (genome-wide DNA methylation) and 87 for the validation analysis (pyrosequencing). Figure 1 and Appendix A display the flow chart of the study and the patient’s characteristics, respectively.

The umbilical cord was clamped and cut upon delivery, and a section of the cord (a piece measuring 1 to 4 cm in length) was immediately stored at −80 °C in RNA later solution. For DNA and RNA extraction, a section of the umbilical cord free of blood vessels and containing Wharton’s Jelly was used in full, without cell type sorting. In this study, we decided to use whole umbilical cord tissue based on the potential clinical application of the results, as this is a spare tissue at birth that is easily available for sample collection, and no extra processing steps are required. Umbilical blood is more difficult to obtain (given the common delay in cord clamping) and also more difficult to process.

### 2.2. Clinical Assessments

Maternal weight and height were measured at each trimester of gestation and before delivery. Gestational age was calculated using the last menstrual period and, whenever possible, was confirmed by ultrasound assessment. Maternal age at conception, height, and pre-pregnancy weight were obtained by questionnaire and crosschecked with clinical records when possible. BMI was calculated as weight divided by height squared (kg/m^2^). Weight gain across trimesters was calculated as follows: 1st-trimester weight gain (difference between 1st-trimester weight and pre-gestational weight), 2nd-trimester weight gain difference between 2nd-trimester weight and 1st-trimester weight) and 3rd-trimester weight gain (difference between 3rd-trimester weight and 2nd-trimester weight). Total GWG was calculated as the difference between the last weight measurement before delivery and pre-pregnancy weight.

Newborn’s weight and length were measured at delivery using a calibrated scale and a measuring board, respectively. Gestational-age- and sex-adjusted standard deviation scores (SDS) were calculated using regional data [25].

At 6 years, weight and height were measured using a calibrated scale and a Harpenden stadiometer, respectively. BMI, age- and sex-adjusted SDS were calculated as mentioned above [25]. The z-score change from weight at birth (BW) to BMI at 6 years (ΔBW − BMI) was calculated as BMI-SDS at 6 yr—Birth weight-SDS. Waist circumference was ascertained in the supine position at the level of the umbilicus. Fat mass (FM) percentage was assessed by bioelectric impedance (Hydra Bioimpedance Analyzer 4200; Xitron Technologies, San Diego CA) using the body weight and lean mass parameters [FM = ((body weight − lean mass)/body weight) × 100] [26]. Carotid intima-media thickness (cIMT) was measured by high-resolution ultrasonography (MyLab™ 25; Esaote, Florence, Italy) as previously described [27]. All ultrasound measurements were performed by the same observer, who was unaware of the clinical and laboratory characteristics of the subjects. The intra-subject coefficient of variation for ultrasound measurements was less than 6%.

### 2.3. DNA Methylome Analysis

Twenty-four women with different degrees of GWG (65% with excessive GWG) were selected for the genome-wide DNA methylation analysis (Figure 1, Appendix A). DNA was extracted from the umbilical cord using the Gentra Pure-Gene Tissue kit (Qiagen, Hilden, Germany) according to the manufacturer’s instructions. DNA methylation profile was performed with the Infinium Human DNA Methylation EPIC 850 K BeadChip array (Illumina Inc., San Diego, CA, USA). DNA was processed, and data were analyzed at Epigenomics Unit and Biostatistics Service from IIS La Fe (Valencia, Spain) as previously described [28]. Briefly, whole genome amplification was done, followed by hybridization on Human Methylation 850 K EPIC BeadChips at 48 °C for 16 h. Subsequently, there was a single nucleotide extension, repeated rounds of staining with antibodies fixed with different fluorophores, and finally, the BeadChip was washed prior to being scanned [28]. Data were quality control pre-processed and normalized by the minfi package (1.26.2 version). Functional normalization (signal background subtraction) to discard probes with a detection *p* > 0.01 and probes that lack signal values in one or more samples and filters (related to sex chromosomes, within SNPs, and multiple homologies) were applied to the raw data. Raw data have been deposited in the Gene Expression Omnibus database (accession number GSE192812). Human gene annotation of each CpG site (CpGs) was performed using the hg19 genome, and RefSeq curated genes [29]. Differential methylation analysis was performed using beta regression models with GWG as a continuous variable, thus avoiding the artificiality of category grouping. Statistical significance for assessing differentially methylated CpGs was set at a false discovery rate (FDR) adjusted *p*-value < 0.01.

Gene ontology (GO) enrichment analysis of the genes annotated by the differentially methylated CpGs was performed with the free software FunRich: Functional Enrichment Analysis Tool (version 3.1.4.) [30,31].

For the validation analysis, CpGs without gene annotation (not available information or genes named orf and LOC) were removed. CpGs were arranged in the order of their odds ratio (OR) value. CpGs that had an OR higher or equal than 1.09 or lower or equal than 0.91 (considered to have a higher biological effect), and another differentially methylated CpGs with the same annotation located within 300 pb of distance (considered to be more relevant), were selected (Figure 1).

### 2.4. Pyrosequencing Analysis

The methylation status of the selected CpGs was validated in 87 subjects (Figure 1, Appendix A) by pyrosequencing bisulfite-treated DNA. Genomic DNA was extracted as described above and 1µg of DNA was bisulfite using the EZ DNA Methylation-Gold kit (Zymo Research, Irvine, CA, USA). Bisulfite-treated DNA (40 ng) was amplified with 0.2 µM of forward and biotinylated-reverse primers (Appendix A). Reactions were conducted in 1X Hot Star Buffer, 0.2 mM dNTPs, and 0.8 U of Hot Star Polymerase (Qiagen, Hilden, Germany) in a total volume of 20 µL. PCR cycling steps were: 15 min at 95 °C followed by four cycles of 20 s at 95 °C, 30 s at 65 °C and 60 s at 72 °C; later four cycles of 20 s at 95 °C, 30 s at 58 °C, 60 s at 72 °C; and 38 cycles of 20 s sat 95 °C, 30 s at the annealing temperature of each pair of primers (Appendix A), 60 s at 72 °C; to finish 3 min at 72 °C. The PCR product was rendered single-stranded and pyrosequenced in a PyroMark Q96 IS (Qiagen, Hilden, Germany) at Instituto de Investigación Biomédica de Málaga (IBIMA, Málaga, Spain) with 4 pmol of the sequencing primer. Raw data were analyzed using the Pyromark CpGs software (Qiagen, Hilden, Germany), and the percentage of methylation for each analyzed CpGs was obtained.

### 2.5. Gene Expression Analysis

Gene expression mRNA levels of the genes annotated by the selected CpGs (*SETD8*, *TMEM214*, *SLIT3*, *RPTOR*, *HOXC8*, and the housekeeping gene *GAPDH*) were measured by RT-qPCR (Taqman Gene Expression assays Hs01029948_m1, Hs00214624_m1, Hs00935843_ms, Hs00375332_m1, Hs00224073_m1, Hs02786624_g1 Thermofisher Scientific, Waltham, MA, USA). Total RNA was extracted and retrotranscribed using the RNeasy mini kit (Qiagen, Hilden, Germany) and the MultiScribe Reverse Transcriptase (Thermofisher Scientific). Reactions were run on a LighCycler480 Real-Time PCR System (Roche Diagnostics, Rotkreuz, Switzerland) using the default cycling conditions. Relative gene expression levels were calculated according to the 2^−ΔCT^ method.

### 2.6. Statistics

Statistical analyses were carried out with SPSS 22.0 (SPSS, Chicago, IL, USA). Results are shown as mean ± standard error of the mean (SEM). Non-normally distributed variables were mathematically transformed to improve symmetry. The difference in means between groups was assessed using Student’s *t*-test and one-way ANOVA. A univariate general linear model to adjust for potential confounders (i.e., maternal pre-gestational BMI and GWG, gestational age, and child’s sex and age) was used. Comparisons of categorical variables were studied using the Chi-square test. Statistical significance was set at *p*-value ≤ 0.05. Accepting an alpha risk of 0.05 and a beta risk of less than 0.2 in a bilateral contrast, a total sample of 111 subjects will allow us to detect significant differences between groups in methylation and gene expression and significant relations thereof with offspring characteristics (GRANMO, IMIM, version 7.12).

## 3. Results

### 3.1. DNA Methylome Analysis

The DNA Methylation array in umbilical tissue samples (*n* = 24) identified 4451 differentially methylated CpGs associated with GWG (adjusted *p*-value < 0.01) that annotated to 2778 RefSeq genes (Figure 1, Appendix A). Concerning CpGs island position, 34% of the differentially methylated CpGs were located in islands, 24% at shore sites, and 8% at shelf sites (Appendix A); concerning gene position, 51% were located in the gene body, and approximately 28% at the transcription start sites (TSS) (Appendix A).

The gene ontology analysis retrieved significant enrichment in biological processes involved in the positive and negative regulation of transcription from RNA polymerase II promoter and ephrin receptor signaling pathway (Appendix A).

Next, we focused on those CpGs (211 CpGs) whose methylation levels were significantly associated with GWG and had another close CpGs (less than 300 pb) annotating for the same gene, as this could reflect biological significance. Among them, the top 12 CpGs most strongly associated with GWG (OR close to 1.1 or 0.91) were chosen for validation. These top 12 CpGs were annotated for five genes: *SETD8* and *TMEM214*, which showed negative associations with GWG; and *SLIT3*, *RPTOR*, and *HOXC8*, which showed positive associations with GWG (Table 1).

### 3.2. Selected CpGs and Association with GWG

The top 12 CpGs, annotating for *SETD8*, *TMEM214*, *HOXC8*, *SLIT3*, and *RPTOR*, were validated in a larger sample (*n* = 87, validation analysis) by bisulfite pyrosequencing. *SETD8*, *TMEM214*, and *HOXC8* CpGs showed very low levels of methylation (mean values for *SETD8*: 0.56 ± 0.08%; *TMEM214*: 2.31 ± 0.27%; *HOXC8*: 1.11 ± 0.11%) while *SLIT3* and *RPTOR* CpGs showed methylation levels around 50% (mean values for *SLIT3*: 61.99 ± 1.04%; *RPTOR*: 38.21 ± 1.16%). *SETD8* (mean expression 0.069 ± 0.005), *TMEM214* (mean expression 0.009 ± 0.001), *HOXC8* (mean expression 0.003 ± 0.001), and *RPTOR* (mean expression 0.021 ± 0.001) genes showed the lowest expression levels and *SLIT3* (mean expression 0.334 ± 0.027) showed the highest expression.

Given that a significant number of subjects (52%) showed 0% methylation for *SETD8*, *HOXC8*, and *TMEM214* CpGs, methylation data were analyzed as qualitative instead of as quantitative data departing from the mean of all CpGs for each given gene and generating two groups of subjects as follows: low methylation levels (those with methylation levels below the sample 50th centile, *n* = 44) and high methylation levels (those with methylation levels above the sample 50th centile, *n* = 43) (Table 2).

No associations were apparent between the methylation levels and total GWG in the validation cohort (Table 2). However, higher methylation of *SETD8* was associated with lower GWG in the 3rd trimester in these subjects (4.09 ± 0.27 vs. 5.75 ± 0.48 kg; *p* < 0.05). In turn, higher methylation of *SLIT3* was associated with higher GWG in the 3rd trimester (5.20 ± 0.40 vs. 4.18 ± 0.30 kg; *p* < 0.05), and higher methylation of *RPTOR* was associated with higher GWG in the 1st trimester (2.28 ± 0.59 vs. 0.84 ± 0.24 kg; *p* < 0.05). These results were consistent with those derived from the screening analysis. *TMEM214* and *HOXC8* methylation groups showed no association with GWG.

### 3.3. CpGs Methylation and Obesity-Related Parameters in the Offspring

We compared the clinical parameters of the offspring at 6 years according to the methylation groups described above (Table 2). Results showed that higher umbilical cord methylation of *SETD8* was related to a higher risk of obesity, including higher BMI-SDS, FM-SDS, ΔBW–BMI, and cIMT (all *p* < 0.05, Figure 2A). Similar results were observed for *RPTOR*, as higher methylation levels in the umbilical cord were associated with higher BMI-SDS and ΔBW–BMI. Conversely, offspring with increased methylation of *SLIT3* had a lower risk of obesity, including lower BMI-SDS, FM-SDS, and ΔBW–BMI (all *p* < 0.05, Figure 2A). *TMEM214* methylation groups were not associated with offspring outcomes, and *HOXC8* methylation groups showed limited associations with offspring BMI-SDS.

### 3.4. Gene Expression and Obesity-Related Parameters in the Offspring

As performed for methylation results, gene expression results were also analyzed as qualitative data generating two groups of subjects as follows: lower gene expression (those with gene expression below the 50th centile; *n* = 30) and higher gene expression (those with gene expression above the 50th centile; *n* = 31) (Table 3).

Inverse associations between methylation levels and gene expression were found for *SETD8* and *SLIT3* (both *p* < 0.05; Table 3). These associations were maintained after adjusting for pregestational BMI, gestational age, and newborn sex in multiple linear regression analyses.

In the mothers, *SETD8* expression levels were inversely associated with gestational BMI and total gestation weight gain (Table 3).

In the offspring, *SETD8* and *SLIT3* expression levels were associated with obesity risk factors at age 6, in opposite directions as compared with their methylation levels (Figure 2B and Table 3). Higher expression of *SETD8* in the umbilical cord was related to lower weight-SDS, BMI-SDS, FM-SDS, waist circumference, and cIMT of the offspring at age 6 years (all *p* < 0.05). Conversely, higher gene expression of *SLIT3* was associated with higher weight-SDS, height-SDS, and FM-SDS (all *p* < 0.05). *RPTOR* expression levels also showed significant associations with obesity-related parameters (ΔBW-BMI and waist circumference), which were in the same direction as those found for methylation analyses (Figure 2B and Table 3). Finally, no, or limited associations were found between *TMEM214* and *HOXC8* gene expression and offspring outcomes.

## 4. Discussion

We have performed a whole methylome study in umbilical cord tissue in relation to maternal GWG status and studied the relationship of the top methylated CpGs and the expression of their annotated genes with the offspring outcomes at 6 years of age. Changes in methylation of three specific genes, namely *SETD8*, *SLIT3*, and *RPTOR* were validated as being associated with maternal GWG. In turn, methylation and gene expression levels of these three genes were associated with the obesity parameters of the offspring at 6 years of age.

Our screening analysis using the Infinium Human DNA Methylation EPIC 850 K BeadChip array showed higher GWG to be associated with differential methylation at 4451 CpGs that annotated to 2778 genes. The gene ontology analysis showed that these genes were involved in transcriptional regulatory pathways. These results add to the evidence supporting the role of the perinatal environment in the modulation of tissue DNA methylation in the newborn [32] and their role in regulating gene expression with potential consequences for the development of offspring obesity in later life [8]. In this sense, Godfrey et al. showed that the methylation status of specific CpGs in the umbilical cord is associated with later adiposity status in the offspring [11].

Among all the genes associated with GWG, we selected those that presented more than one differentially methylated CpGs and undertook validation of the top 12 CpGs (annotating for five genes) using pyrosequencing. Among these, changes in methylation of 3 specific genes (*SETD8*, *SLIT3*, and *RPTOR*) were validated to be associated with maternal GWG. In turn, methylation and gene expression levels of these same genes were associated with obesity parameters in the offspring. Importantly, the association of gene expression with offspring outcomes paralleled the association of methylation levels of the same three genes with maternal GWG, suggesting that methylation of the specific genes could explain, at least in part, the relationship between GWG and obesity risk of the offspring in later life. Moreover, both *SETD8* and *SLIT3* showed an inverse association between methylation and expression levels, indicating a possible regulation of gene expression by methylation. This observation also explains why we observed opposite associations between methylation and gene expression levels for *SETD8* and *SLIT3* and offspring obesity-related parameters.

SETD8 is a histone lysine methyltransferase involved in transcriptional regulation, cell cycle progression, DNA replication, and damage repair [33]. The methylation of lysine residues in histones is an important epigenetic event; hence, SETD8 is indeed an epigenetic regulator. Our data showed that *SETD8* methylation negatively associates with *SETD8* gene expression supporting previous findings in the literature, which postulates that methylation of CpGs at transcription start sites usually downregulates gene expression [34,35,36]. Moreover, *SETD8* expression levels are associated with a lower risk of obesity in the offspring at age 6 years, in the opposite direction to their methylation levels. Although there are no previous studies on *SETD8* methylation, its gene expression has been broadly studied [37]. Recent evidence has shown that SETD8 plays a pivotal role in the regulation of peroxisome proliferator-activated receptor (PPARγ) expression and in adipogenesis [37,38]. Li et al. also showed that SETD8 is a Wnt signaling mediator that regulates the transcription of Wnt-activated genes [39]. Wnt signaling plays important roles in several human metabolic diseases, including obesity [40]. In agreement with these data, our results highlight a possible role of umbilical cord SETD8 in the regulation of obesity-related pathways.

SLIT3 belongs to the Slit-Robo protein family of Robo ligands. Robo receptors are transmembrane proteins involved in cell signaling, and slit-Robo unions are related to cell proliferation, stem cell regulation, angiogenesis, and organ development [41]. There are few studies on *SLIT3* methylation and none in umbilical cord tissue; however, Lim et al. found that *SLIT3* differential methylated CpGs (DMC) were altered in the placenta from pregnant women with preeclampsia, demonstrating that its methylation may be affected by the gestational environment [42]. Our results showed that *SLIT3* methylation relates to a worse metabolic phenotype in the offspring, also in the opposite direction to their methylation levels. No previous studies have related SLIT3 with obesity; nevertheless, Lim et al. suggest a pro-inflammatory role of SLIT3 in amnion and myometrial tissues [43]. Obesity has consistently been linked to an inflammatory state, and taken together, these results evoke a possible function of SLIT3 in obesity developmental mechanisms.

RPTOR, the regulatory-associated protein of mTOR, is known to play a role in lipogenesis by controlling mTORC1 activity [44] and regulates cell growth in response to nutrient and insulin levels [45]. Our results showed that *RPTOR* hypermethylation in the umbilical cord correlated with increased BMI-SDS and ΔBW–BMI in offspring at age 6 years. *RPTOR* has previously been directly associated with obesity in several genome-wide association studies (GWAS) [44,46] and epigenome-wide association studies (EWAS) [47].

The design of the current study is subject to limitations. The primary limitation is that we studied only umbilical cord tissue, and in future studies, it would be interesting to analyze other tissues. Secondly, cell heterogeneity in the umbilical cord tissue was not taken into account, given the wide variety of cell types in this tissue, and therefore similar studies to elucidate differences in the methylation between each cell type are needed. However, the use of whole umbilical cord tissue would ease the implementation of this analysis at a clinical level, as no previous cell sorting would be required. Thirdly, our participants decreased by 30% on the follow-up; conversely, having a follow-up rate of 70% may also be seen as a strength in longitudinal studies. The strengths of our study also include the longitudinal design and the parallel analysis of methylation and gene expression. It would be interesting to repeat this study in additional cohorts, thus increasing the significance and relevance of our results and drawing more definitive conclusions.

## 5. Conclusions

In summary, our work set down that GWG is associated with a specific pattern of umbilical cord tissue methylation and, in turn, these methylation changes, as well as gene expression levels, are associated with offspring obesity-related parameters at age 6 years. The identification of epigenetic markers associated with obesity-related outcomes that are measured early in life may provide insight into the developmental origins of metabolic diseases. Most importantly, these marks could be used to identify children at risk of obesity and to design personalized interventions in order to prevent this condition, which could be translated into a reduction in adult obesity and its prevalent comorbidities.

## Figures and Tables

**Figure 1 nutrients-15-03175-f001:**
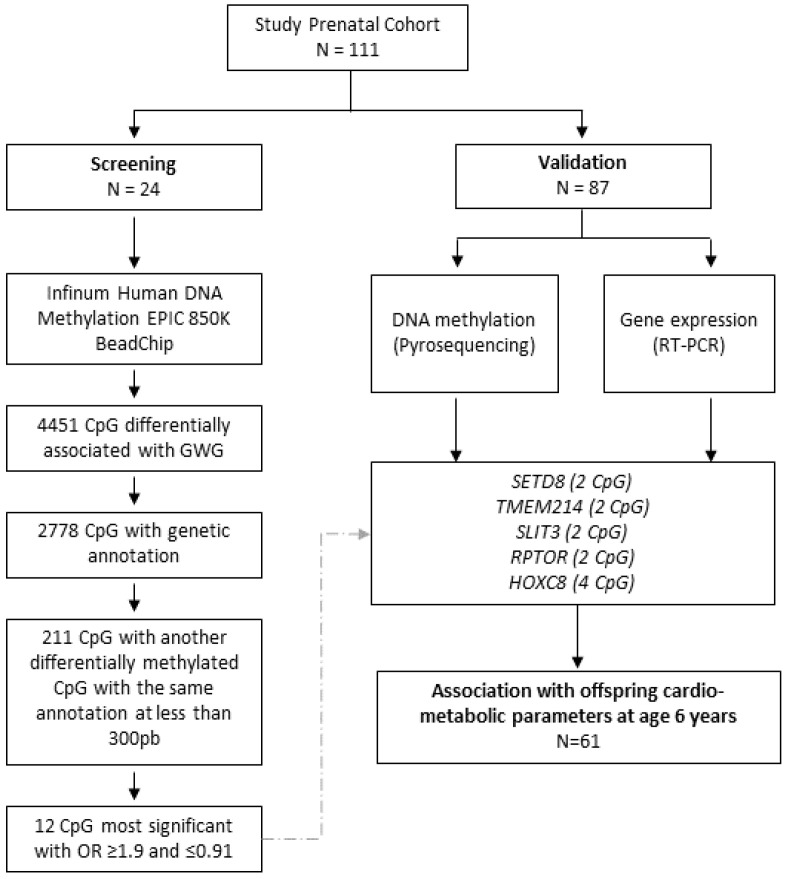
Flow chart of the methodology followed in the study.

**Figure 2 nutrients-15-03175-f002:**
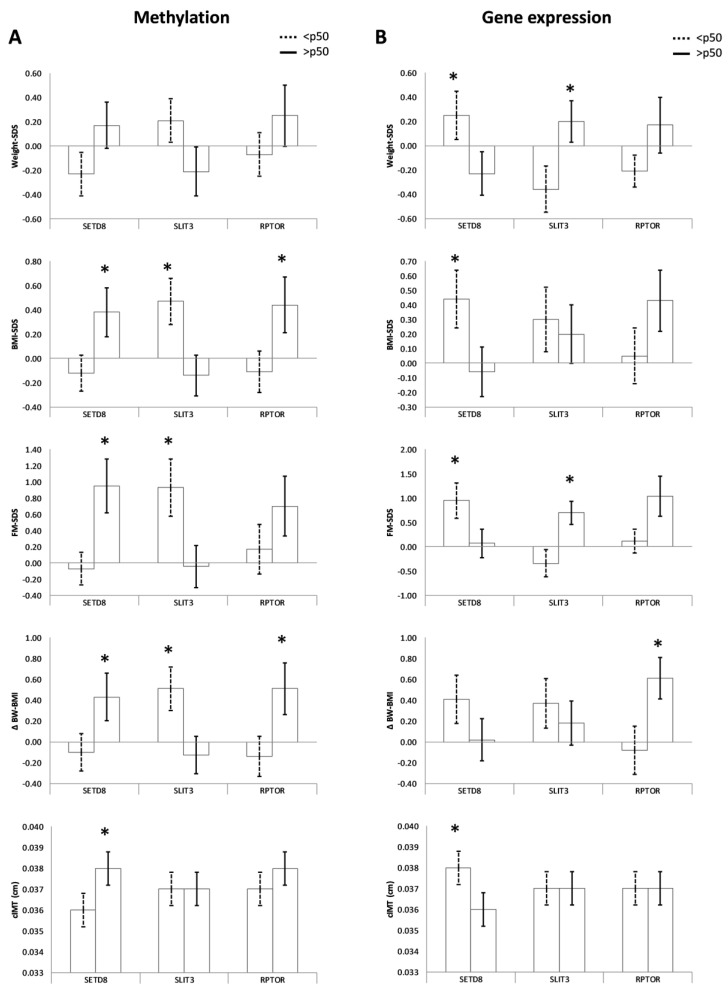
Relationship of DNA methylation (**A**) and gene expression (**B**) with offspring parameters at age 6 years. * *p* ≤ 0.05 for differences in univariate general linear models after adjusting for potential confounders (maternal pregestational BMI and GWG, gestational age at birth, and child’s sex and age). BMI: Body mass index; FM: Fat mass; cIMT: Carotid intima-media thickness.

**Table 1 nutrients-15-03175-t001:** CpG sites chosen for validation.

Gene		Estimate Coefficient	OR	Chromosome	Position	Relation to Gene	Relation to CpG Island
*SETD8*	CpG 1	−0.10000675	0.90483131	12	123868662	TSS200	Island
	CpG 2	−0.04700338	0.95408418	12	123868665	TSS200	Island
*TMEM214*	CpG 1	−0.05779695	0.94384157	2	27255615	TSS200	Island
	CpG 2	−0.08774743	0.9159922	2	27255618	TSS200	Island
*SLIT3*	CpG 1	0.07056568	1.07311505	5	168271855	Body	NA
	CpG 2	0.10201274	1.10739758	5	168271859	Body	NA
*RPTOR*	CpG 1	0.08914043	1.09323417	17	78915842	Body	Island
	CpG 2	0.07583486	1.07878441	17	78915881	Body	Island
*HOXC8*	CpG 1	0.0872253	1.09114248	12	54402697	TSS200	Island
	CpG 2	0.121894	1.12963435	12	54402699	TSS200	Island
	CpG 3	0.11646746	1.12352096	12	54402714	TSS200	Island
	CpG 4	0.08129304	1.0846887	12	54402717	TSS200	Island

**Table 2 nutrients-15-03175-t002:** Comparison of the clinical parameters according to methylation groups.

	Methylation of *SETD8*		Methylation of *SLIT3*		Methylation of *RPTOR*	
	<50th Centile	>50th Centile	<50th Centile	>50th Centile	<50th Centile	>50th Centile
**Mother**						
Age (years)	30.53 ± 0.65	31.07 ± 0.56	30.87 ± 0.55	30.71 ± 0.80	30.21 ± 0.70	30.97 ± 0.63
Pregestational BMI	24.33 ± 0.66	24.61 ± 0.64	24.44 ± 0.62	24.43 ± 0.74	24.54 ± 0.70	24.29 ± 0.47
1st-trimester BMI	24.74 ± 0.66	25.54 ± 0.62	25.16 ± 0.64	25.01 ± 0.71	24.94 ± 0.70	25.08 ± 0.64
2nd-trimester BMI	26.61 ± 0.59	27.83 ± 0.62	27.33 ± 0.62	27.44 ± 0.66	26.95 ± 0.61	27.14 ± 0.68
3rd-trimester BMI	29.03 ± 0.63	29.35 ± 0.60	28.91 ± 0.65	29.42 ± 0.67	29.10 ± 0.66	29.05 ± 0.66
1st-trimester GWG (kg)	1.24 ± 0.25	2.01 ± 0.53	1.47 ± 0.31	1.95 ± 0.56	0.84 ± 0.24	2.28 ± 0.59 *
2nd-trimester GWG (kg)	6.04 ± 0.45	6.17 ± 0.50	6.19 ± 0.47	6.14 ± 0.53	6.08 ± 0.50	5.88 ± 0.44
3rd-trimester GWG (kg)	5.75 ± 0.48	4.09 ± 0.27 *	4.18 ± 0.30	5.20 ± 0.40 *	5.07 ± 0.44	5.02 ± 0.46
Total GWG (kg)	14.40 ± 0.73	14.23 ± 0.87	13.51 ± 0.73	14.74 ± 0.85	13.45 ± 0.75	14.91 ± 0.86
**Newborn**						
Gender (%F)	47	53	47	52	50	50
GA (wk)	39.82 ± 0.16	39.76 ± 0.16	39.79 ± 0.15	39.87 ± 0.18	39.67 ± 0.16	39.87 ± 0.18
Placental weight (kg)	5.86 ± 0.16	6.09 ± 0.18	5.74 ± 0.14	6.19 ± 0.21	5.82 ± 0.14	6.05 ± 0.21
Birth weight-SDS (z-score)	−0.01 ± 0.09	−0.01 ± 0.09	−0.06 ± 0.09	0.01 ± 0.10	−0.03 ± 0.10	−0.04 ± 0.09
Birth length-SDS (z-score)	−0.25 ± 0.11	−0.18 ± 0.16	−0.14 ± 0.13	−0.34 ± 0.16	−0.25 ± 0.14	−0.31 ± 0.14
**Child**						
Gender (%F)	42	58	58	42	44	56
Age (years)	5.87 ± 0.17	5.79 ± 0.18	5.80 ± 0.18	5.76 ± 0.19	5.71 ± 0.19	5.78 ± 0.18
Weight-SDS (z-score)	−0.23 ± 0.18	0.17 ± 0.19	0.21 ± 0.18	−0.21 ± 0.20	−0.07 ± 0.18	0.25 ± 0.25
Height-SDS (z-score)	−0.08 ± 0.20	−0.17 ± 0.24	−0.04 ± 0.19	−0.24 ± 0.28	0.05 ± 0.24	−0.21 ± 0.25
BMI-SDS (z-score)	−0.12 ± 0.15	0.38 ± 0.20 *	0.47 ± 0.19	−0.14 ± 0.17 *	−0.11 ± 0.17	0.44 ± 0.23 *
FM-SDS (z-score)	−0.07 ± 0.20	0.95 ± 0.33 *	0.93 ± 0.35	−0.04 ± 0.26 *	0.17 ± 0.31	0.70 ± 0.37
∆ BW − BMI (z-score)	−0.10 ± 0.18	0.43 ± 0.23 *	0.51 ± 0.21	−0.13 ± 0.18 *	−0.14 ± 0.19	0.51 ± 0.25 *
Waist (cm)	56.32 ± 1.15	57.48 ± 1.53	58.00 ± 1.36	55.65 ± 1.52	56.08 ± 1.27	56.64 ± 1.39
cIMT (cm)	0.036 ± 0.01	0.038 ± 0.01 *	0.037 ± 0.001	0.037 ± 0.001	0.037 ± 0.01	0.038 ± 0.01

Data are expressed as the mean of all the CpG methylation ± SEM. BMI: Body-mass index; 1: 1st trimester; 2: 2nd trimester; 3: 3rd trimester; GWG: gestational weight gain; GA: gestational age; SDS: Standard-deviation score; FM: fat mass; ∆ BW − BMI: BMI-SDS at 6 yr—Birthweight-SDS; cIMT: carotid intima-media thickness. * *p* < 0.05 for differences in univariate general linear models after adjusting for potential confounders (maternal pregestational BMI and GWG, gestational age, and child’s sex and age).

**Table 3 nutrients-15-03175-t003:** Comparison of the clinical parameters according to gene expression groups.

	Relative Expression *SETD8*		Relative Expression *SLIT3*		Relative Expression *RPTOR*	
	<50th Centile	>50th Centile	<50th Centile	>50th Centile	<50th Centile	>50th Centile
**Mother**						
Age (yrs)	31.39 ± 0.61	30.19 ± 0.60	30.55 ± 0.72	31.05 ± 0.48	29.93 ± 0.63	31.67 ± 0.56 *
Pregestational BMI	24.90 ± 0.60	24.01 ± 0.68	24.56 ± 0.66	24.37 ± 0.63	24.61 ± 0.62	24.31 ± 0.67
1st trimester BMI	25.68 ± 0.61	24.55 ± 0.67	25.22 ± 0.64	25.04 ± 0.65	25.19 ± 0.63	25.06 ± 0.67
2nd trimester BMI	28.18 ± 0.61	26.23 ± 0.58 *	27.32 ± 0.61	27.09 ± 0.62	26.97 ± 0.58	27.45 ± 0.64
3rd trimester BMI	30.09 ± 0.59	28.24 ± 0.61 *	29.25 ± 0.60	29.13 ± 0.63	29.30 ± 0.58	29.08 ± 0.65
1st-trimester GWG (kg)	1.61 ± 0.26	1.62 ± 0.53	1.90 ± 0.51	1.33 ± 0.27	1.37 ± 0.24	1.88 ± 0.55
2nd-trimester GWG (kg)	6.70 ± 0.52	5.47 ± 0.39	5.95 ± 0.51	6.26 ± 0.44	5.93 ± 0.48	6.28 ± 0.47
3rd-trimester GWG (kg)	5.16 ± 0.48	4.71 ± 0.33	5.19 ± 0.40	4.67 ± 0.42	5.51 ± 0.45	4.3 ± 0.35 *
Total GWG (kg)	15.63 ± 0.82	12.97 ± 0.72 *	14.17 ± 0.78	14.45 ± 0.82	14.50 ± 0.76	14.12 ± 0.84
**Newborn**						
Gender (%F)	37	63 *	50	50	55	45
GA (wk)	39.73 ± 0.18	39.86 ± 0.14	39.95 ± 0.15	39.63 ± 0.16	39.95 ± 0.16	39.63 ± 0.15
Placental weight (kg)	6.02 ± 0.16	5.93 ± 0.18	6.08 ± 17.31	5.86 ± 17.86	6.10 ± 0.16	5.83 ± 0.19
Weight-SDS (z-score)	0.07 ± 0.07	−0.10 ± 0.10	−0.02 ± 0.09	−0.01 ± 0.09	0.14 ± 0.08	−0.17 ± 0.09 *
Length-SDS (z-score)	−0.26 ± 0.13	−0.17 ± 0.14	−0.31 ± 0.14	−0.12 ± 0.13	−0.09 ± 0.13	−0.35 ± 0.14
**Child**						
Gender (%F)	41	59	41	59	56	44
Age (yrs)	5.88 ± 0.17	5.77 ± 0.17	5.80 ± 0.18	5.82 ± 0.16	5.61 ± 0.16	6.01 ± 0.16
Weight-SDS (z-score)	0.25 ± 0.20	−0.23 ± 0.18 *	−0.36 ± 0.19	0.20 ± 0.17 *	−0.21 ± 0.13	0.17 ± 0.23
Height-SDS (z-score)	0.05 ± 0.21	−0.36 ± 0.22	−0.43 ± 0.27	0.12 ± 0.16 *	−0.12 ± 0.20	−0.09 ± 0.22
BMI-SDS (z-score)	0.44 ± 0.20	−0.06 ± 0.17 *	0.30 ± 0.22	0.20 ± 0.20	0.05 ± 0.19	0.43 ± 0.21
FM-SDS (z-score)	0.95 ± 0.36	0.07 ± 0.29 *	−0.34 ± 0.28	0.70 ± 0.24 *	0.12 ± 0.24	1.04 ± 0.41
∆ BW-BMI (z-score)	0.41 ± 0.23	0.02 ± 0.20	0.37 ± 0.24	0.18 ± 0.21	−0.08 ± 0.23	0.61 ± 0.20 *
Waist (cm)	59.46 ± 1.40	55.25 ± 1.25 *	56.56 ± 1.20	57.75 ± 1.40	55.42 ± 1.03	59.11 ± 1.54 *
cIMT (cm)	0.038 ± 0.001	0.036 ± 0.001 *	0.037 ± 0.001	0.037 ± 0.001	0.037 ± 0.001	0.037 ± 0.001
**Umbilical cord**						
Methylation (%)	0.66 ± 0.14	0.39 ± 0.07 *	65.18 ± 1.43	58.80 ± 1.36 *	38.92 ± 1.49	37.49 ± 1.79

Data are expressed as mean ± SEM. BMI: Body mass index; GWG: gestational weight gain; GA: gestational age; SDS: Standard-deviation score; FM: fat mass; ∆ BW-BMI: BMI-SDS at 6 yr—Birthweight-SDS; cIMT: carotid intima-media thickness. * *p* < 0.05 for differences in univariate general linear models after adjusting for potential confounders (maternal pregestational BMI and GWG, gestational age and child’s sex and age).

## Data Availability

Raw data have been deposited in the Gene Expression Omnibus database (accession number GSE192812).

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
