# Peer review of "Gestational Weight Gain Relates to DNA Methylation in Umbilical Cord, Which, In Turn, Associates with Offspring Obesity-Related Parameters"

_nutrients, 2023, doi:10.3390/nu15143175_

Round 1
Reviewer 1 Report
This study evaluated the association between gestational weight gain and DNA methylation in the umbilical cord tissue and evaluated whether DNA methylation is associated with obesity related parameters in the offspring at 6 years of age. It is an interesting manuscript.
The authors should explain whether this is a prospective study or a retrospective analysis of prospectively collected data.
It is not clear when during gestation were the participating women recruited?
Is the sample of 111 women big enough to draw any definite conclusions?
In the introduction section the authors should further explain why umbilical cord tissue was used in this study.
Possible clinical implications of this study should be explained in the conclusions section.
Author Response
Thank you for giving us the opportunity to submit a revised version of our manuscript. We appreciate the time and effort that you have dedicated to providing your valuable feedback on our manuscript. We have incorporated the changes to reflect the suggestions provided.
------------------------------
Response to Reviewer 1
Thank you for giving us the opportunity to submit a revised version of our manuscript. We appreciate the time and effort that you have dedicated to providing your valuable feedback on our manuscript. We have incorporated the changes to reflect the suggestions provided. Please see our detailed responses below.
This study evaluated the association between gestational weight gain and DNA
methylation in the umbilical cord tissue and evaluated whether DNA methylation is associated with obesity related parameters in the offspring at 6 years of age. It is an interesting manuscript.
The authors should explain whether this is a prospective study or a retrospective
analysis of prospectively collected data.
Thank you. This is a prospective longitudinal study of 111 pregnant Caucasian women and their newborns, who were included in a mother-children cohort in Girona, north-eastern Spain. We have added this information to the “subjects and samples” section.
It is not clear when during gestation were the participating women recruited?
Thank you. Women were recruited in a prenatal primary care setting during the first trimester of gestation. We have added this information to the “subjects and samples” section.
Is the sample of 111 women big enough to draw any definite conclusions?
Thank you. Accepting an alpha risk of 0.05 and a beta risk of less than 0.2 in a bilateral contrast, a total sample of 111 subjects will allow us to detect significant differences between groups in methylation and gene expression and significant relations thereof with offspring characteristics (GRANMO, IMIM, version 7.12). We have added this information to the “statistics” section. Moreover, at the end of the discussion we have stated that it would be interesting to repeat this study in additional cohorts, thus increasing the significance and relevance of our results and to draw more definitive conclusions.
In the introduction section the authors should further explain why umbilical cord tissue was used in this study.
In this study we decided to use whole umbilical cord tissue based on the potential clinical application of the results, as this is a spare tissue at birth that is easily available for sample collection and no extra processing steps are required. Umbilical blood is more difficult to obtain (given the common delay in cord clamping) and also more difficult to process. We have added this information to the “methods” section.
Possible clinical implications of this study should be explained in the conclusions section.
The identification of epigenetic markers associated with obesity-related outcomes that are measured early in life may provide insight into the developmental origins of metabolic diseases. Most importantly, these marks could be used to identify children at risk of obesity and to design personalized interventions in order to prevent this condition, which could be translated into a reduction of adult obesity and its prevalent comorbidities. We have added this information to the “conclusions” section.

Reviewer 2 Report
Mas-Pares et al. examined, in umbilical cord tissue, the association between DNA methylation patterns related to gestational weight gain and offspring cardio-metabolic outcomes. Overall, the study is well designed and the outcomes are appropriate.
Only a few minor concerns:
A few format issues:
“ There is a convincing association between pre and postnatal environmental exposures and disease risk in later life.[1], [2]” should be written as ”There is a convincing association between pre and postnatal environmental exposures and disease risk in later life [1-2].”
Reference format seems inadequate.
A few typos: line 310, be et al. instead of et al.,:
Author Response
Thank you for giving us the opportunity to submit a revised version of our manuscript. We appreciate the time and effort that you have dedicated to providing your valuable feedback on our manuscript. We have incorporated the changes to reflect the suggestions provided.
---------------------------
Response to Reviewer 2
Thank you for giving us the opportunity to submit a revised version of our manuscript. We appreciate the time and effort that you have dedicated to providing your valuable feedback on our manuscript. We have incorporated the changes to reflect the suggestions provided. Please see our detailed responses below.
Mas-Pares et al. examined, in umbilical cord tissue, the association between DNA
methylation patterns related to gestational weight gain and offspring cardio-metabolic outcomes. Overall, the study is well designed and the outcomes are appropriate. Only a few minor concerns:
A few format issues:
“ There is a convincing association between pre and postnatal environmental exposures and disease risk in later life.[1], [2]” should be written as ”There is a convincing association between pre and postnatal environmental exposures and disease risk in later life [1-2].” Reference format seems inadequate.
Thank you. We have corrected the reference format across all the manuscript.
A few typos: line 310, be et al. instead of et al.,:
Thank you. We have revised the manuscript and corrected the typo errors.
